

# Rediscovery and redescription of the only known mosasaur bone from the Turonian (Upper Cretaceous) of Poland

Tomasz Skawiński

Department of Palaeozoology, Faculty of Biological Sciences, University of Wroclaw, Wrocław, Poland

## ABSTRACT

Mosasaur remains from Poland are very rare and are restricted mostly to the Campanian and Maastrichtian. The only currently known pre-Campanian records come from the Turonian strata in the Opole area, southwestern Poland. One of them is a single tooth which probably belongs to a yaguarasaurine while the other is an incomplete vertebra, for many years considered lost. The latter specimen has recently been found and is redescribed in this article. Its most characteristic feature is a strong dorsoventral compression of the articular surfaces. This is similar to the condition observed in basal mosasauroids such as halisaurines and tethysaurines. Unfortunately, due to its incompleteness, the rediscovered specimen cannot be confidently referred to any of these clades and can only be described as a probable non-mosasaurine, non-plioplatecarpine, non-tylosaurine mosasauroid. Despite its uncertain phylogenetic position, it is important from a historical point of view and as only the second record (and the only bone record) of mosasauroids from the Turonian of Poland.

# INTRODUCTION

Mosasaurs (Mosasauroidea) are one of the major groups of Mesozoic marine reptiles. This species-rich clade of predatory aquatic squamates has a rich fossil record spanning over 30 million years, from the early Cenomanian, some 98 million years ago, to the end of the Cretaceous, 66 million years ago. The Cenomanian mosasaur fossils are rare and the group became more diverse during the next geological age, the Turonian (*e.g.*, *Polcyn et al., 2014*). Therefore, Turonian fossils can give us more information about diversification of this important group which is still not fully understood (*e.g.*, *Madzia & Cau, 2017*; *Simões et al., 2017*).

Turonian tetrapod remains from Poland are very rare and currently known only from a few sites in the Opole area (southwestern Poland). They were recently reviewed by *Sachs et al. (2018)* who described a few polycotylid plesiosaur teeth, an unidentified plesiosaur limb bone and a probable russellosaurinan mosasaur tooth (*Sachs et al., 2018*). Most of these specimens were discovered already in the second half of the 19th century and were first described by *Leonhard (1897)*. One of them was a postcranial bone discovered by Schumann, an officer at the Ministry of Defence (*Sachs et al., 2018*). This fossil was originally identified as a plesiosaur phalanx by *Leonhard (1897)*. He described it as 'Plesiosauridarum

Corresponding author
Tomasz Skawiński,
tomasz.skawinski@uwr.edu.pl

gen.' which can be translated as 'plesiosaurid genus'. *Sachs et al. (2018)* were unable to locate this specimen but based on the illustration provided by *Leonhard (1897)*, reidentified it as a damaged mosasauroid vertebra. Recently, I found this fossil in the collection of the Department of Palaeozoology, University of Wrocław. Here, I attempt to provide a redescription of the specimen and discuss its potential systematic position.

## MATERIALS & METHODS

### Geological settings

Unfortunately, the exact locality data are not available for ZPALUWr/R133. However, they may be approximated based on circumstantial evidence. The specimen was collected by Schumann from the Turonian strata at Opole (Fig. 1). He discovered another fossil (a polycotylid tooth ZPALUWr/R245) from the Turonian at Opole. It may be presumed that these two fossils were collected at the same site. If so, both these specimens are probably not older than the zone UC7 (*sensu Burnett, 1998*) as indicated by the calcareous nannoplankton data (*Sachs et al., 2018*). In addition, all currently known marine amniote remains from the Opole Trough (reviewed by *Sachs et al., 2018*) are most probably early Turonian in age. Therefore, it is most parsimonious to assume that ZPALUWr/R133 is also the same age.

### Measurements and image acquisition

The measurements were made to the nearest 0.01 mm using a Mitutoyo digital caliper. Each of the distances was measured three times and a mean of them represents the value reported below.

The photographs were taken using a Canon EOS 90D digital camera. A photogrammetric three-dimensional model (File S1) was generated using Meshroom (AliceVision) and MeshLab (*Cignoni et al., 2008*).

## RESULTS

### Systematic Palaeontology

Squamata *Oppel (1811)*
Mosasauroidea *Gervais (1853)*
?Mosasauridae *Gervais (1853)*

### Material

ZPALUWr/R133, an isolated, incomplete dorsal vertebra (Fig. 2).

### Locality and age

Opole, southwestern Poland; most likely early Turonian, Late Cretaceous (see *Sachs et al., 2018*).

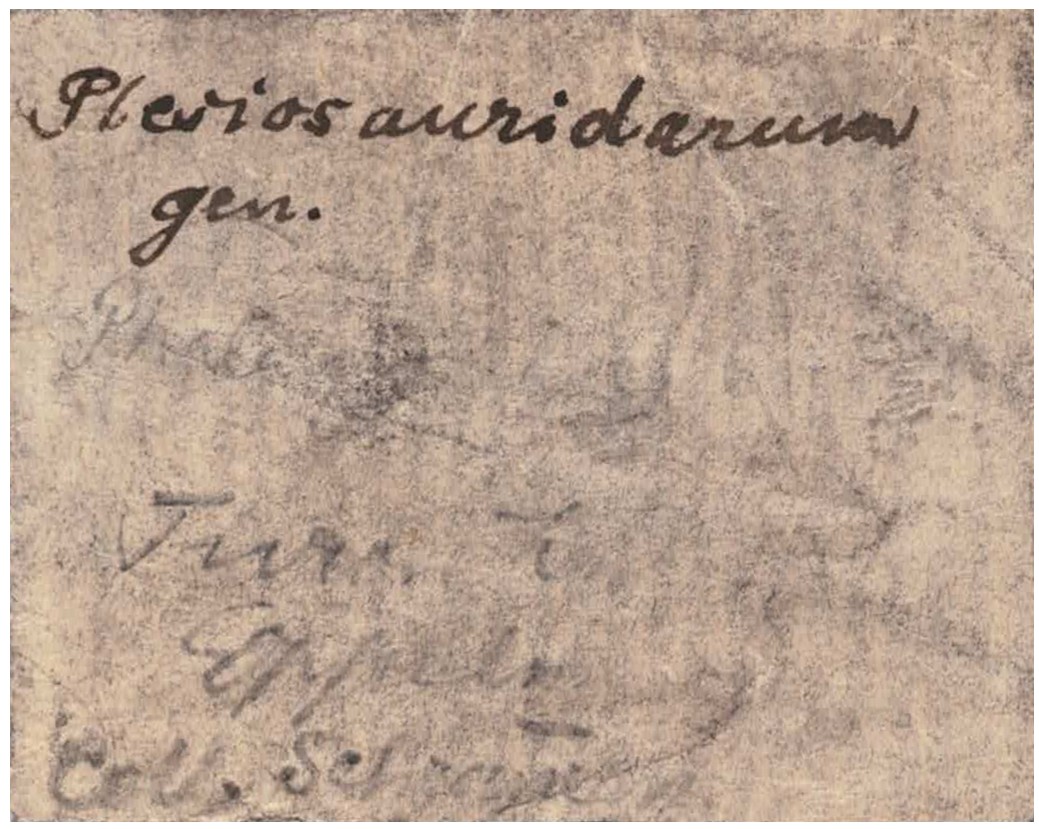

**Figure 1** **An old label associated with ZPALUWr/R133.** The inscription reads: "Plesiosauridarum gen. Phalanx. Turon. Oppeln. Coll. Schumann".

## Morphological description

The rediscovered specimen (ZPALUWr/R133) is a damaged vertebral centrum (Fig. 2). It is procoelus and dorsoventrally compressed. The state of preservation is the poorest in the anterior part of the vertebra, where it is damaged on both the dorsal (which is the base of the vertebral canal) and ventral sides. Only the base of the right synapophysis is preserved. The borders of the anterior cotyle are poorly preserved so its exact size and shape are difficult to reconstruct. However, it seems that its width was noticeably greater than its height. This matches the dimensions of the condyle which is also ellipsoidal, being much wider than high. In lateral view, the condyle is straight rather than inclined. There are faint but clear longitudinal ridges on the ventral side of the centrum just anterior to the condyle (in places where the surface is least eroded). The lateral margins are mostly damaged but they appear to be complete at the posterior right side; this indicates that the precondylar constriction was absent or only minimal. There is no sign of a hypapophysis. The ventral surface of the centrum is noticeably concave in lateral view. All other parts of the vertebra are not preserved.

The length of the centrum along the midline is 79.04 mm. The condyle is 24.91 mm high and 44.52 mm wide in its highest and widest points, respectively. This gives a height/width

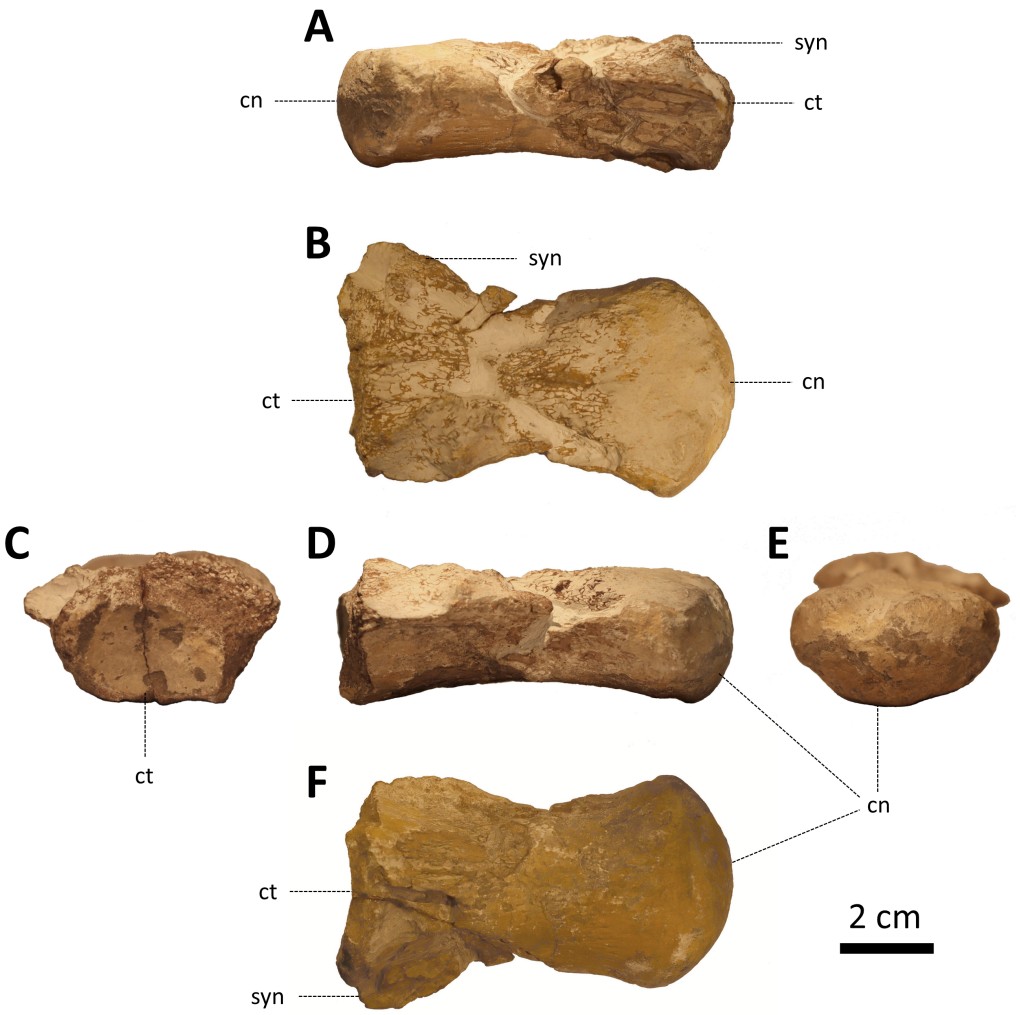

**Figure 2** **ZPALUWr/R133, a damaged mosasaur vertebral centrum from the Turonian of the Opole area, Poland.** (A) Right lateral view, (B) dorsal view, (C) anterior view, (D) left lateral view, (E) posterior view, (F) ventral view. Scale bar = 2 cm. Abbreviations: cn, condyle; ct, cotyle; syn, synapophysis.

ratio of ~0.60. The cotyle is broken; the preserved part is approximately 18 mm high and 28 mm wide.

## COMPARISONS

The vertebra is poorly preserved and most of the structures enabling precise anatomical and taxonomical identification are missing. The lack of a hypapophysis indicates that it most probably represents a dorsal rather than cervical vertebra. The only well-preserved part is the condyle which is much wider than high. This distinguishes ZPALUWr/R133 from mosasaurines, tylosaurines and plioplatecarpines in which dorsal vertebrae have much more circular condyles. Such condition is present *e.g.*, in mosasaurines *Dallasaurus* (*Bell & Polcyn, 2005*), *Mosasaurus* (*e.g.*, *Houssaye, 2008*), *Plotosaurus* (*e.g.*, *Lindgren, Caldwell & Jagt, 2008*), *Clidastes* and *Globidens* (*e.g.*, *Russell, 1967*), tylosaurines *Tylosaurus* and

*Hainosaurus* (*e.g.*, *Russell, 1967*; *Jiménez-Huidobro & Caldwell, 2016*) and plioplatecarpines *Platecarpus* and *Plioplatecarpus* (*e.g.*, *Russell, 1967*; *Mulder, 2003*). In its proportions, ZPALUWr/R133 is more similar to halisaurines, *Haasiasaurus* and non-tylosaurine, non-plioplatecarpine representatives of Russellosaurina which show much more dorsoventrally compressed condyle. Such compression is present in the halisaurine *Halisaurus* (*e.g.*, *Russell, 1967*; *Holmes & Sues, 2000*; *Mulder, 2003*; *Bardet et al., 2005*), *Haasiasaurus* (*Houssaye, 2008*), tethysaurines *Tethysaurus* (*Bardet, Pereda Suberbiola & Jalil, 2003*) and *Pannoniasaurus* (*Makádi, Caldwell & Osi, 2012*) and yaguarasaurine *Romeosaurus* (*Palci, Caldwell & Papazzoni, 2013*). Dorsoventral compression of the condyle is also present in a basal mosasauroid *Komensaurus* (*Caldwell & Palci, 2007*). Unfortunately, other basal mosasauroids ('aigialosaurs') cannot be directly compared to ZPALUWr/R133 in this respect.

The lack of precondylar constriction is common in dorsal vertebrae in basal mosasauroids (*Sato et al., 2018*) but differentiates ZPALUWr/R133 from *Pannoniasaurus* (*Makádi, Caldwell & Osi, 2012*), *Portunatasaurus* (*Campbell Mekarski et al., 2019*) and OTBE Obr-3609, a mosasauroid from the Campanian of Hokkaido, Japan, in which the constriction is present (*Sato et al., 2018*).

The condyle is not tilted posterodorsally, similarly to *Tethysaurus* (*Bardet, Pereda Suberbiola & Jalil, 2003*), but in contrast to *Halisaurus* (*Holmes & Sues, 2000*), *Romeosaurus* (*Palci, Caldwell & Papazzoni, 2013*) and *Pannoniasaurus* (*Makádi, Caldwell & Osi, 2012*).

The condyle is vertical in ZPALUWr/R133, unlike in most other basal mosasauroids (OTBE Obr-3609, *Komensaurus*, *Haasiasaurus*, *Halisaurus*, *Pannoniasaurus*, *Tethysaurus*; *e.g.*, *Dutchak & Caldwell, 2009*; *Sato et al., 2018*), in which it is inclined, but similarly to *Dallasaurus* (*Bell & Polcyn, 2005*) and *Aigialosaurus dalmaticus* (as coded by *Dutchak & Caldwell, 2009*).

ZPALUWr/R133 is much larger than all vertebrae of *Halisaurus* listed or figured by *Holmes & Sues (2000)* and *Bardet et al. (2005)*. However, a large size was attained by at least some halisaurines as shown by *Pluridens serpentis* which is estimated at 6–10 m in length (*Longrich et al., 2021*). ZPALUWr/R133 probably represents an animal larger than *Tethysaurus* (*Bardet, Pereda Suberbiola & Jalil, 2003*) and *Romeosaurus* (*Palci, Caldwell & Papazzoni, 2013*) but smaller or comparable to the largest described individuals of *Pannoniasaurus*, estimated at 6 m in length (*Makádi, Caldwell & Osi, 2012*).

## DISCUSSION

The most characteristic and arguably the most informative feature of the redescribed vertebra is its strong dorsoventral compression. Therefore, the question of whether this trait is a result of a taphonomic flattening of the specimen is justified. The answer cannot be conclusive given the fact that only a single, incomplete bone is known. However, even though the bone is incomplete and damaged, the shape and proportions of certain structures do not seem to be distorted. Moreover, a similar degree of the compression of articular surfaces is observed in numerous basal mosasauroids (see 'Comparisons') and occurs commonly in varanoid squamates (*e.g.*, *Holmes & Sues, 2000*). If it is indeed an

 

ancestral condition for mosasauroids, its presence would not be unexpected in a geologically relatively old (early Turonian) taxon.

Establishing the phylogenetic position of ZPALUWr/R133 is difficult. If the dorsoventral compression of the articular surfaces is a genuine feature—as argued here—the specimen shows the greatest similarities to basal mosasauroids, *i.e.*, non-mosasaurine, non-platecarpine and non-tylosaurine taxa. This is expected given its Turonian age. The degree of the condylar compression and the lack of precondylar constriction are similar to the tethysaurine *Tethysaurus* and halisaurine *Halisaurus*. In the latter, however, the condyle is somewhat tilted dorsally (*Holmes & Sues, 2000*), unlike ZPALUWr/R133. On the other hand, the condyles in *Tethysaurus* dorsal vertebrae are oblique (*Bardet, Pereda Suberbiola & Jalil, 2003*), in contrast to the Opole mosasauroid. In *Halisaurus*, the condyles are less obliquely oriented (*Holmes & Sues, 2000*) and thus more similar to ZPALUWr/R133. In light of these data, it seems that of currently known mosasauroid vertebrae, those of *Halisaurus* and *Tethysaurus* are most similar, albeit none of them is a perfect match. Obviously, taxonomic identification of isolated bones, especially as incomplete and damaged as ZPALUWr/R133, must be taken with caution.

The fossil record of non-tylosaurine and non-plioplatecarpine russellosaurinans extends at least to early Turonian. Fossils of these mosasaurs from this time are currently known from Europe, Africa, North America and South America but are more widespread in Laurasian continents (*e.g.*, *Polcyn et al., 2008*; *Jiménez-Huidobro, Simões & Caldwell, 2017*). The yaguarasaurines are represented by *Romeosaurus sorbinii* from the lower-middle Turonian and *R. fumanensis* from the middle Turonian-early Santonian of Italy (*Palci, Caldwell & Papazzoni, 2013*), *Russellosaurus coheni* from the middle Turonian of Texas (*Polcyn & Bell, 2005*), *Yaguarasaurus columbianus* from the upper Turonian-lower Coniacian of Colombia (*Jiménez-Huidobro, Simões & Caldwell, 2017*), BMB 007158 (holotype of '*Mosasaurus gracilis*') from the middle Turonian of England (*Street & Caldwell, 2014*) and MUZ-299, a right surangular from Mexico. The latter specimen may come from the lower Turonian or upper Cenomanian (*Jiménez-Huidobro et al., 2021*). Importantly, an isolated tooth referred tentatively to Yaguarasaurinae is known from probably the same locality as ZPALUWr/R133 (*Sachs et al., 2018*). Turonian tethysaurines are less diverse. *Tethysaurus* fossils were discovered in lower Turonian strata in Morocco (*Bardet, Pereda Suberbiola & Jalil, 2003*) but the second currently named taxon, *Pannoniasaurus*, is only known from the Santonian (*Makádi, Caldwell & Osi, 2012*). It is important to note that a slightly younger (probably middle or late Turonian) tethysaurine maxilla was described from a geographically close location in the Bohemian Cretaceous Basin (*Kear et al., 2014*). Putative tethysaurine teeth are also known from the upper Turonian of Austria (*Osi et al., 2019*). In contrast to russellosaurinans, the halisaurine fossil record is currently restricted to post-Coniacian strata (*Caldwell & Bell, 1995*; *Bardet et al., 2005*; *Polcyn et al., 2012*). However, according to current phylogenetic hypotheses (*Madzia & Cau, 2017*; *Simões et al., 2017*), it must extend at least to early Turonian, so the presence of Turonian fossils of halisaurines would not be unexpected.

The relationships within the Mosasauroidea are still not fully resolved (*Madzia & Cau, 2017*; *Simões et al., 2017*). Even if the similarities between ZPALUWr/R133 and *Tethysaurus*

reflect their relatively close phylogenetic relationship, this does not necessarily indicate that the former belongs to the more inclusive clade Mosasauridae, as tethysaurines and yaguarasaurines are positioned as non-mosasaurid mosasauroids in some analyses (*e.g.*, unweighted parsimony analysis in *Madzia & Cau, 2017*: Fig. 2).

The reanalysis of ZPALUWr/R133 has implications for marine tetrapod diversity in the Turonian strata of the Opole area. The number of hitherto discovered fossils is very low and indicates the presence of polycotylid plesiosaurs and probably a basal russellosaurinan mosasaur, possibly related to yaguarasaurines (*Sachs et al., 2018*). Except for these few fragmentary remains, the putative turtle remains were mentioned (*Jagt-Yazykova & Jagt, 2015*) but have not been formally described. Although ZPALUWr/R133 seems to be most similar to halisaurines and tethysaurines, it cannot be excluded that it is conspecific with the putative yaguarasaurine from Opole, represented by an isolated tooth crown (ZPALUWr/R248; *Sachs et al., 2018*). Unfortunately, our knowledge of the postcranial anatomy of yaguarasaurines is incomplete, so a detailed comparison cannot be made. However, *Romeosaurus* differs from ZPALUWr/R133 in the dorsal tilt of the vertebral condyle (*Palci, Caldwell & Papazzoni, 2013*).

The fossil record of mosasaurs in Poland is scarce and limited mostly to Campanian and Maastrichtian forms. It includes fossils referred to *Mosasaurus* cf. *hoffmani* and *M.* cf. *lemonnieri* (*Sulimski, 1968*; *Machalski et al., 2003*), *Dollosaurus* cf. *lutugini* (*Machalski et al., 2003*; *Hornung, Reich & Frerichs, 2018*) and two species of *Hainosaurus* (*Machalski et al., 2003*; *Jagt et al., 2005*). The only pre-Campanian records are the isolated tooth crown ZPALUWr/R248 (*Leonhard, 1897*; *Sachs et al., 2018*) and an incomplete vertebra ZPALUWr/R133 (see above) from the Turonian of Opole. Mosasaur remains from the Turonian are also rare in Poland's neighbouring countries; they are currently not known from Germany (*Sachs, Hornung & Reich, 2015*) and a tethysaurine maxilla as well as an indeterminate mosasaur tooth were described from the Czech Republic (*Ekrt et al., 2001*; *Kear et al., 2014*). This is not surprising because the Turonian was still a relatively early period in mosasaur evolution, though over ten Turonian forms are currently known (*e.g.*, *Polcyn et al., 2014*).

## CONCLUSIONS

ZPALUWr/R133, an isolated mosasauroid vertebra, originally described by *Leonhard (1897)* from the Turonian strata at Opole, has been rediscovered and redescribed. As the specimen lacks a hypapophysis, it most probably represents a dorsal vertebra. Its most characteristic feature is a strong dorsoventral compression which corresponds well with the condition observed in basal taxa such as halisaurines, yaguarasaurines and tethysaurines. However, its incompleteness prevents a confident referral to any of these clades. Thus, ZPALUWr/R133 can be described as probably representing a non-mosasaurine, non-plioplatecarpine, non-tylosaurine mosasauroid. It is only the second record (and the only known bone record) of mosasauroids from the Turonian of Poland.

### Institutional Abbreviations

| | |
|---|---|
| **BMB** | Booth Museum of Natural History, Brighton, United Kingdom |
| **MUZ** | Museo Paleontológico de Múzquiz, Melchor Múzquiz, Coahuila, Mexico |
| **OTBE** | Obira Town Board of Education, Obira, Japan |
| **ZPALUWr** | Department of Palaeozoology, Faculty of Biological Sciences, University of Wrocław, Wrocław, Poland |

## ACKNOWLEDGEMENTS

I thank Bartosz Borczyk and Aleksandra Kropczyk (University of Wrocław) for their technical help. Constructive comments made by Daniel Madzia and Michael Polcyn greatly improved the manuscript. I thank Mark Young for handling the review process.

### Funding

The author received no funding for this work. The funders had no role in study design, data collection and analysis, decision to publish, or preparation of the manuscript.

### Competing Interests

The author declares that he has no competing interests.

### Author Contributions

- Tomasz Skawiński conceived and designed the experiments, performed the experiments, analyzed the data, prepared figures and/or tables, authored or reviewed drafts of the article, and approved the final draft.

### Data Availability

The three-dimensional model of ZPALUWr/R133 (File S1) is available in STL and PLY formats in Figshare: Skawiński, Tomasz (2022): ZPALUWr R133. figshare. Media. Available at https://doi.org/10.6084/m9.figshare.21215816.v2.

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
