# Peer review of "Rediscovery and redescription of the only known mosasaur bone from the Turonian (Upper Cretaceous) of Poland"

_PeerJ, doi:10.7717/peerj.14278_

## Round 0.1 · original submission · Minor Revisions

Dear authors,

Apologies for the delay in returning this review. Based on the reviewers' comments I have accepted their decision of 'minor revisions'.

I look forward to receiving your revised manuscript.

·

Basic reporting

Thank you for the opportunity to review the MS by Tomasz Skawiński titled "Rediscovery and redescription of the only known mosasaur bone from the Turonian (Upper Cretaceous) of Poland".

The text is clear and well-written and the structure of the MS is logical. Figures are both clear though I would probably suggest to annotate particular structural elements that are discussed in the main text.

Experimental design

The research is certainly original and perfectly within the scope of the journal. The research question is well-defined and rigorous investigation has been performed to a high technical and ethical standard.

Validity of the findings

All underlying data have been provided. Conclusions are well-stated.

Additional comments

It's great that the author rediscovered the specimen and offered his time to provide such detailed description.

Nothing would have really happened if the MS was published as is. I have just some very minor comments that I included in the annotated PDF.

I look very much forward to seeing the MS published

·

Basic reporting

no comment

Experimental design

no comment

Validity of the findings

no comment

Additional comments

To the author and editors,
I have reviewed the manuscript “Rediscovery and redescription of the only known mosasaur bone from the Turonian (Upper Cretaceous) of Poland” by Tomasz Skawiński and recommend acceptance with only a few minor comments. The contribution is quite straight forward and does a good job documenting the rediscovery of a historically important specimen and in providing an updated description and figure. It would be good to have a 3d model (3d pdf?) of the vertebra as a supplement if that is possible. The comparisons could be improved by including more detailed metric analysis such as that by Caldwell and Bell (1995; their figures 3 and 4), though that study was limited to cervical vertebrae. I do understand the difficulty in data collection for such an undertaking, so I leave it to the authors to determine if that is worth adding to this contribution. It may also be worth expanding the discussion on the stratigraphic and geographic distribution russellosaurians and halisaurines to narrow possible taxonomic assignment. The record of russellosaurians clearly extends to the Lower Turonian in North America, North Africa, and Europe (Polcyn et al., 2008; 2014; Bardet et al., 2003; Sachs et al, 2018). Halisaurines are currently known only from the Santonian-Maastrichtian. Otherwise, the writing is solid and professional, the only suggestion would be to not use the unnecessary adverb “very” (e.g. very rare; very low; very incomplete; very scarce; very surprising).
Sincerely,
Michael J. Polcyn


Bardet, N., Suberbiola, X.P. and Jalil, N.E., 2003. A new mosasauroid (Squamata) from the Late Cretaceous (Turonian) of Morocco. Comptes Rendus Palevol, 2(8), pp.607-616.
Caldwell, M.W. and Bell Jr, G.L., 1995. Halisaurus sp.(Mosasauridae) from the Upper Cretaceous (? Santonian) of east-central Peru, and the taxonomic utility of mosasaur cervical vertebrae. Journal of Vertebrate Paleontology, 15(3), pp.532-544.
Polcyn, M.J., Bell Jr, G.L., Shimada, K. and Everhart, M.J., 2008. The oldest North American mosasaurs (Squamata: Mosasauridae) from the Turonian (Upper Cretaceous) of Kansas and Texas with comments on the radiation of major mosasaur clades. In Proceedings of the Second Mosasaur Meeting, Fort Hays Studies Special (No. 3, pp. 137-155).
Polcyn, M.J., Jacobs, L.L., Araújo, R., Schulp, A.S. and Mateus, O., 2014. Physical drivers of mosasaur evolution. Palaeogeography, Palaeoclimatology, Palaeoecology, 400, pp.17-27.
Sachs, S., Jagt, J.W., Niedźwiedzki, R., Kędzierski, M., Jagt-Yazykova, E.A. and Kear, B.P., 2018. Turonian marine amniotes from the Opole area in southwest Poland. Cretaceous Research, 84, pp.578-587.

---

## Round 0.2 · accepted · Accept

Dear authors,

Based on your rebuttal letter, I have accepted your manuscript for publication.

In due course the production team will be in touch to take you through the proofing stages.

Congratulations again, and I hope you will choose PeerJ as your publication venue in the future.